# Blind-Watermarking—Proof-of-Concept of a Novel Approach to Ensure Batch Traceability for 3D Printed Tablets

**DOI:** 10.3390/pharmaceutics14020432

**Published:** 2022-02-17

**Authors:** Hellen Windolf, Rebecca Chamberlain, Arnaud Delmotte, Julian Quodbach

**Affiliations:** 1Institute of Pharmaceutics and Biopharmaceutics, Heinrich Heine University, Universitätsstr. 1, 40225 Düsseldorf, Germany; hellen.windolf@hhu.de (H.W.); rebecca.chamberlain@hhu.de (R.C.); 2Optical Media Interface Laboratory, Nara Institute of Science and Technology, 8916-5 Takayama-cho, Nara 630-0192, Japan; delmotte_ar@yahoo.fr; 3Department of Pharmaceutical Sciences, Utrecht Institute for Pharmaceutical Sciences, Utrecht University, Universiteitsweg 99, 3584 CG Utrecht, The Netherlands

**Keywords:** FDM 3D printing, traceability, blind-watermarking, anti-counterfeiting, falsified medicine, personalized medicine

## Abstract

Falsified medicines are a major issue and a threat around the world. Various approaches are currently being investigated to mitigate the threat. In this study, a concept is tested that encodes binary digits (bits) on the surface of Fused Deposition Modelling (FDM) 3D printed geometries. All that is needed is a computer, a FDM 3D printer and a paper scanner for detection. For the experiments, eleven different formulations were tested, covering the most used polymers for 3D printing in pharma: Ethylene-vinyl acetate (EVA), polyvinyl alcohol (PVA), polylactic acid (PLA), Hypromellose (HPMC), ethyl cellulose (EC), basic butylated-methacrylate-copolymer (EPO), and ammonio-methacrylate-copolymer type A (ERL). In addition, the scanning process and printing process were evaluated. It was possible to print up to 32 bits per side on oblong shaped tablets corresponding to the dimensions of market preparations of oblong tablets and capsules. Not all polymers or polymer blends were suitable for this method. Only PVA, PLA, EC, EC+HPMC, and EPO allowed the detection of bits with the scanner. EVA and ERL had too much surface roughness, too low viscosity, and cooled down too slowly preventing the detection of bits. It was observed that the addition of a colorant or active pharmaceutical ingredient (API) could facilitate the detection process. Thus, the process could be transferred for 3D printed pharmaceuticals, but further improvement is necessary to increase robustness and allow use for more materials.

## 1. Introduction

A global threat to healthcare is falsified and substandard medicine. Worldwide, an estimated 10% of medicines on the market are falsified [1,2,3,4,5,6,7,8]. In developing countries, the percentage of falsified and substandard drugs is higher, at about 10–30% [3,9,10,11,12,13,14,15,16]. Particularly at risk of counterfeiting are those drugs that are expensive or promise high sales. In developing countries, these are often antibiotics, viral drugs, or malaria preparations [14,15,16,17]. In rich countries, falsified medicines of new and expensive so-called “lifestyle pharmaceuticals” are most common, for example, hormones, steroids, and antihistamines [18]. Falsified drugs usually contain no active pharmaceutical ingredient (API), the wrong API or the wrong amount of API, and thus lead to no effect in the best cases [19]. However, they can also cause allergies and other side effects or even death [17,20]. For example, falsified vaccines do not contain an effective component and cannot protect patients from disease [21]. Counterfeited and falsified pharmaceuticals and medical devices are also currently a major issue during the SARS-CoV-2 pandemic. Falsified COVID-19 vaccines, fake masks, hand sanitizers, and self-test kits are sold to private persons, hospitals, and community pharmacies [22,23,24,25]. To prevent and trace these crimes, various systems have been integrated and are further developed [17,26,27,28,29,30]. Since February 2019, for EU Member States, pharmaceuticals must correspond to the requirements of the Commission Delegated Regulation 2016/161 [31] and Directive 2011/62/EU [32]. Pharmaceuticals marketed in the EU must be serialized and equipped with tamper-evident or tamper-resistant function (manipulation-safe sealing of packaging carton) [33]. This sealing contains a unique 2D barcode which includes the batch, serial and national identification number, expiry date, and product information [34]. In addition, it is sometimes possible to identify falsified medicines visually (e.g., packaging, labelling, dosage units, watermarks, holograms), physically (discoloration, microscopical investigations of the surface, evaluation of disintegration), or chemically (API assay via spectroscopic, spectrometric, or chromatographic measurements) [30,35].

However, a new traceability-system is needed for personalized medicine, which is not or will likely not be produced industrially on a large scale, but individually in compounding centers, community pharmacies, or hospitals in small or on-demand batches [36,37,38,39,40,41]. These tailored medicines would be produced in the absence of serialized/anti-counterfeit packaging. Therefore, it would be highly advantageous if the traceability system is directly included in or on the dosage form. For oral dispersible films (ODF), Edinger et al. investigated a QR Code traceability system, which is printed with an ink-jet printer on a previous manufactured ODF [42]. In another study, a laser-based technology was used to mark an individual QR code on the surface of a tablet [30]. Rui et al. ink-jet printed fingerprint characters on the surface of tablets, which are detectable by pictures with regular smartphones [43].

Currently, a lot of research is being done on 3D printed dosage forms, as they enable low-cost, personalized drug therapy, especially the fused deposition modelling (FDM) method [37,44,45,46,47,48,49,50,51,52]. With this technique, a drug-loaded filament is conveyed through a heated nozzle on a print bed and the previous designed object is built layer-by-layer. The required filament is previously manufactured via hot-melt extrusion (HME). This type of individual dosage form is particularly interesting for developing countries. Production is inexpensive and the dosage can be flexibly adjusted so that a larger part of the population can be supplied with medicine. This could counteract the circumvention of the health care system and the purchase of drugs on the black market. However, commercial FDM 3D printers are available for anyone to purchase, and the process is easy to learn, so counterfeiting can be expected with this innovative dosage form as well. That is why various research groups are currently working on different ways to avoid counterfeit. Trenfield et al. developed a track-and trace system for 3D printed oral dosage forms with a combined 2D printing technology for printed QR codes and data matrices on the surface of “printlets” [39]. It was possible to scan these codes with a smartphone device.

In this proof-of-concept study, the blind-watermarking concept developed by Delmotte et al. [53] was transferred to FDM 3D printed oral dosage forms. In this method, binary digits (bits) are inserted on the flat sides of the object via a variation of the layer thickness. The bits are inserted into the previously created G-Code using a self-programmed C++ script. The insertion of individual blind-watermarking bits is intended to implement a security system that will prevent falsifying the drug. The change of the layer thickness has no influence on printing time, appearance, weight, or API content of the dosage form. In addition, no other equipment is needed for the implementation, except for a FDM 3D printer. A simple paper scanner is used to detect the bits as well as a Python script. The concept was tested by the research group on large non-pharmaceutical objects printed with polylactic acid (PLA). This traceability approach could improve the safety of 3D printed tablets, as the process could be established in community pharmacies and hospitals, requiring no equipment other than a FDM 3D printer and a paper scanner. In our study, oblong shaped tablets were designed, and it was examined whether bits could also be inserted on these geometries. Various materials were tested that could be considered for FDM printed oral dosage forms, in some cases also containing API. Different dimensions, variable number of bits, scanning methods, and two layer heights were tested.

## 2. Materials and Methods

### 2.1. Materials

The transfer of the blind-watermarking concept of Delmotte et al. [53] was first tested with commercial polylactic acid filaments (PLA, Bavaria filaments, Freilassing, Germany). After the geometries and G-Codes were created with the desired number of bits, self-extruded pharmaceutical filament compositions were tested (Table 1). These filaments differed in appearance, in color, in roughness, and in their melt viscosity.

### 2.2. Methods

#### 2.2.1. Hot Melt Extrusion

The self-extruded filaments were prepared by hot-melt extrusion (HME) [37,44,52,54,55]. A co-rotating twin-screw extruder (Pharmalab HME 16; Thermo Fisher Scientific, Waltham, MA, USA) was used with an in-house manufactured die (1.85 mm diameter) to produce filaments with a diameter of 1.75 mm. A haul-off unit of a winder (Model 846700, Brabender, Duisburg, Germany) was used to achieve the required filament diameter. This was controlled with a laser-based diameter measurement module (Laser 2025 T, Sikora, Bremen, Germany) with a readout rate of 1 Hz.

#### 2.2.2. Creation of Geometries, G-Codes and Bits

Different oblong tablets were designed with various lengths and heights to investigate what sizes are necessary for a certain number of bits and how many bits can fit on a large oblong tablet. Since bits are generated only on straight, flat sides and can thus be scanned with a 2D scanner, the oblong design was selected. For the design, the computer-aided design (CAD) software Fusion360^®^ (Autodesk, San Rafael, USA) was chosen. For generating the G-Code, PrusaSlicer^®^ (2.2.0, Prusa Research, Prague, Czech Republic) was used. In the settings, the temperature, speed, and layer height were adjusted (Section 2.2.4). The extrusion width was set to 0.4 mm and the variable layer thickness option disabled, so the C++ script for bit-generation could insert the layer thickness changes to encode the bits. In addition, care was taken to ensure that each new layer starts at the same position and the resulting seam is not in the watermarked patch. In the C++ script for bit-insertion, the length of the flat tablet-side for the bits was set (Figure 1, green + red) as well as the number of bits per line, number of bits in height, and number of parity bits. Subsequently, a G-Code with the desired number of bits and parity-bits was created and inserted in the G-Code of the tablet geometry.

#### 2.2.3. Watermark Embedding

For the blind-watermarking method, the layer thickness is locally modified, which results in a pattern on the surface of the 3D printed object. Normally, the layer thickness of a FDM 3D print is constant with little noise. For embedding the watermark, a number of bits is selected, as well as the number of separating layers between the bits. The code is formed by the interaction of two layers, which in sum always have the same thickness. If the lower layer becomes thinner (1 − a) to encode a 0, the upper layer compensates this with 1 + a layer thickness. If the lower layer is thicker with 1 + a, to encode a 1, the upper layer balances this again with 1 − a layer thickness. Thus, despite the differences in thickness between the layers above and below, the result is an even layer so that the code is clearly recognizable (Figure 2). As a minimum distance between encoding bit layers, two separating layers were selected. It was avoided to insert the blind-watermark too close to the bottom or top of the tablet, because the tablets often stuck to the print bed or because the printing precision was insufficient in these layers. Therefore, the bits were only encoded at least four layers above the print bed and at least four layers below the top of the tablet.

#### 2.2.4. 3D Printing Process

The oblong tablets were printed with a FDM 3D Printer Prusa i3MK3 (Prusa Research, Prague, Czech Republic). The settings of the printing process were adjusted for each filament (Table 2). Print settings were determined manually to enable the best possible print image. In most cases, the best results were achieved at the lowest possible temperatures. If the printing temperature was too low, the nozzle became clogged, and no molten filament flowed through it. If the temperature was too high, the surface of the tablet became uneven. For EVA and PVA/PVA + PZQ filaments, higher temperatures had to be used because the layers adhered poorly to each other. At higher temperatures, they were better bonded. EVA filaments were very flexible, and the conveying wheels in the print head could not be used fully to transport the filament through the nozzle, otherwise the filament would wrap around the conveying wheels after a few minutes. The necessary transport to the nozzle could only be ensured by a high printing temperature, as this caused the filament to melt faster and offer less resistance. For a fast cooling of the printed object, the fan was activated. The printing speed was set to 10 mm/s, which is very low for FDM 3D printing. However, for the materials used and following the recommendation of Delmotte et al. [53], a low printing speed should reduce artifacts.

#### 2.2.5. Scan and Detection

To detect the bits, the printed oblong tablets (*n* = 3) were placed with the flat side on a paper scanner (Epson Expression Premium XP-610, Suwa, Nagano Prefecture, Japan) and scanned with the parameters shown in Table 3. The resolution was set to 1200 dpi. Higher resolution settings did not result in better scans and detectability but increased the processing time of the analytical computer script. During scanning, the tablets were covered with a black box so that the process would not be disturbed by room light and the scanning light would not be reflected.

Afterwards, the scanned files were analyzed with a self-programmed Python script for detection. A region of interest surrounding the watermark area is defined and the script runs an algorithm to extract the encoded bits by determination of layer thickness variations. A more detailed description of the algorithm can be found in Delmotte et al. [53]. Afterwards the result is displayed (Figure 3).

#### 2.2.6. Melt Viscosity Measurements

To be able to describe the print behavior of the polymers and blends used, the rheological properties were investigated (*n* = 1). The viscosity was measured with a Modular Advanced Rheometer System (HAAKE MARS 60, Thermo Fisher Scientific, Waltham, MA, USA). Samples of 500 mg each were weighed in. The gap was adjusted to 1 mm and an angular speed of 6.3 rad/s was set. A temperature range was scanned to be able to follow the viscosity curve of the polymers. This range covered the print temperature used. The data was recorded with HAAKE RheoWin (4.87.0006, Thermo Fisher, Waltham, MA, USA) with a frequency of 1 Hz. For the measurements, the API was replaced with mannitol to reduce the toxicity profile of the mixtures.

## 3. Results and Discussion

### 3.1. Minimum and Maximum Size of Oblong Shaped Tablets

The blind-watermarking system can, in its current version, only modify straight and flat surfaces. No bits can be implemented on rounded surfaces. Additionally, the long side must have a certain length so that the script can recognize the side and insert a desired number of bits. As a security measure, parity bits are inserted to recognize errors in the detected bits. The minimum bit count is four bits per site (2 × 2) with four parity bits (Table 4). In the original publication [53], a layer height of 0.2 mm was recommended. With this most simple setup, 16 different combinations are possible (2^4^) for information deposit, and the required length was calculated to be 12 mm (Figure 4).

Subsequently, it was determined how many bits fit on an oblong shaped tablet of maximum size, which will still meet the criteria for swallowability. For this purpose, market preparations with large oblong tablets or capsules were examined regarding their size and the dimensions were adopted [57]. The largest oblong tablets have a length of 23 mm, a height of 6 mm and a width of 8 mm (e.g., Amoxicillin Sandoz 1000 mg: 23 mm × 8 mm × 6 mm; Furobeta^®^ 500 betapharm Arzneimittel GmbH: 23 × 8 × 6 mm; Rosuvastatin/Amlodipin-ratiopharm^®^ capsules 23 × 8.1 mm). A dosage form of such dimensions was designed (Figure 5). The number of bits was increased until the patch became too large so that the required distance between bits was no longer possible. The maximum number of bits per line was nine bits (including one parity bit per row) and five bits in height, again with one parity bit per column (Table 5).

This number of bits generates a wide spectrum of possible combinations (2^32^). Thus, this method allows variable information content from the smallest bit set of 4 bits to 32 bits per side. Depending on the size of the tablet, the number of bits can be adjusted for the necessary information content.

### 3.2. Variation in Layer Height

To further increase the number of bits per tablet side, it was tested whether it is possible to reduce the layer height from 0.2 mm as recommended by Delmotte et al. [53] to 0.1 mm, thus doubling the number of bits per side. The smallest tablet size (12 × 4 × 4 mm) was chosen for this purpose, as it is particularly interesting to be able to increase the bit number for smaller dimensions (Figure 6). The filament used was PVA + colorant.

With the smaller layer height, in theory up to 18 bits can be encoded in the given geometry. The detection for the reference tablet with 0.2 mm layer height worked without problems, and the correct code could be detected directly (Figure 7 top). For the tablet with 0.1 mm layer height, both patches could be detected, but the correct code was not generated immediately. The correct code was determined by the error correction function shown in Figure 7 (bottom) using the parity bits. The lower distance between encoding layers made detection of the bits difficult. It is possible to decrease the layer height and thus increase the number of bits per tablet. Yet, a layer height of 0.1 mm should be considered to be too low. Proper detection using the parity bits is feasible, but considering that the reduction of the layer height from 0.2 mm to 0.1 mm also doubles the printing time per tablet, the drawbacks become prohibitive.

### 3.3. Optimization of the Scanning Process

During the scanning process it was observed that there is a difference in the visibility of bits, depending on the orientation of the tablet to the scan light.

When the samples were oriented perpendicular to the moving scan light, individual bits reflected the light, and the bits were better visible on the image than in a parallel orientation to the scan light. In parallel orientation, hardly any contrast was visible between the bit and the encoding layer. Yet, the layer structure itself was better visible (Figure 8, used bit code Table 6).

For an automatized detection, the parallel orientation to the direction of the scanning light is more suitable, as the layer structure can be seen more clearly and the differences in layer thickness can be detected more easily. The script is oriented on the layer pattern that results from the 0.2 mm layer heights. The bit variations due to (1 − a) and (1 + a) layer thicknesses are recognized by the script due to the deviation from the normal 0.2 mm layer pattern. This is better detectable with the parallel scan. For the eye, on the other hand, the blind watermarking code is better detectable with the perpendicular alignment, since the reflections of the bits clearly show the intended unevenness. Here, however, the layer structure is not clearly visible.

### 3.4. Variations of the Material

To test the applicability of blind-watermarking for 3D printed tablets, various materials were evaluated. Common polymers that are often used for FDM 3D printing of drugs were utilized: PLA, PVA, EPO, EVA, EC, HPMC, PVP-VA, and ERL [37,47,52,55,58,59,60,61,62,63]. These polymers were in some cases mixed with APIs, but also with colorants to observe any influence. The same bit code was used as shown in Table 4 and Figure 4, as well as the associated G-codes with the appropriate print temperatures for each filament. 13 × 5 × 5 mm tablets of each formulation were printed and scanned and the detection was evaluated. Only formulations where the bits were identified directly in all cases were considered suitable (Table 7). A layer height of 0.2 mm was used to keep the error level low and to be able to draw more accurate conclusions about the materials. The tablets were scanned in both parallel and perpendicular orientations to obtain optimal images suitable for the script. In the end, it was always the images in parallel orientation that had the most suitable structure for the script.

During printing and scanning, it became apparent that not every material is suitable for a blind-watermarking approach. The printed tablets made of PLA (Table 7 (a)) could be printed and scanned well. The bits were recognized by the written Python script without any problems. Delmotte et al. [53] already investigated different colors of commercial PLA filaments. By adjusting the brightness and contrasts during the scanning process, different colors could be used for blind-watermarking. In some printed tablets (Table 7 (b–d)), despite adjusting the brightness and contrast of the scanner, the visibility of the bits could not be ensured because the dosage forms were too transparent and reflective (PVA, PVA + PZQ, PVA + PDM) and the scan light was reflected by the object. For the eye, the bits were visible, but not detectable with the scanning light. Using a different imaging technique might be able to solve this issue. In our study, we examined if this issue can be solved by mixing a colorant into the filament so that it was more opaque (PVA + Methylene blue, Table 7 (e)) with less reflection. The reflection-problem should also be solved by adding an API or excipient that did not completely dissolve in the polymer or melted during HME and 3D printing, so that the filament looked slightly milky. As a result, the scanning light was not reflected as strongly, and the bits could be detected (PVA + Triam, Table 7 (f)). The printed tablets with EPO, EC and EC + HPMC (Table 7 (g–i)) were able to visualize the bits, so that the bits could be presented well during the scanning process and the detection script could read out the code. The tablets had a cloudy appearance and neither color nor transparency had a negative impact.

EVA filaments did not result in visible bits imprinted into the tablets (Table 7 (j + k)). It was assumed that the formulations with EVA had a too low melt viscosity. In this case the polymer would only flow out of the nozzle and not retain the bit structure. This could not be remedied by lowering the print temperature from 220 °C to 210 °C, because the nozzle would clog, and the flexible filament would begin to wrap around the conveyor wheels in the print head. During printing it was observed that the printed filament is also likely to take longer to cool and solidify so that the movement of the print head and the following layers destroy the fine bit structure.

The detection of tablets printed from filaments with a rough surface after HME, which is also apparent after printing (EVA + PVP-VA + API; ERL + API, Figure 7 (k + l)) was not possible. A rough surface can result from immiscible excipients and APIs in HME, high polymer blend viscosity, as well as from a high proportion of unmelted, suspended components [64,65]. This noise of the rough surface makes it impossible to recognize the inserted bits.

During the printing process, it was also observed that setting the correct print temperature had a major impact on the appearance of blind-watermarking patches. In addition to lowering the melt viscosity, increased temperature can also lead to the formation of gas bubbles. In the case of PVA + PZQ and EPO+API filaments, even a small difference in printing temperature (2–9 °C) resulted in significant changes in the printed material (Figure 9). If the temperature was slightly too high, gas bubbles were formed in the printed filament, which made the surface of the tablet appear inhomogeneous and did not allow bits to be detected. This may be due to thermal degradation of the API or polymer, moisture in the filament, or due to the release of water. Since PVA is a very hydrophilic polymer, it may absorb water from the environment after a short storage time, which evaporates during printing and leaves gas bubbles. The API PDM contains hydrate water, which is degraded at high temperature [52]. These processes can be controlled by adjusting the print temperature and optimizing the storage conditions. Since lowering the printing temperature increases the melt viscosity, this can cause the nozzle of the 3D printer to clog, and the filament cannot be printed. This must be considered when selecting the polymer composition.

Since it was assumed that the melt viscosity seems to have a major influence on the printing result, rheological measurements of the formulations without API were conducted to find a possible quantifiable parameter to exclude or include filaments in advance for this process (Table 8). The filament blends containing ERL were omitted, as it was the roughness that led to the poor detection and not the melt viscosity.

The results of the viscosity measurement confirm that the melt viscosity has a major influence on the blind-watermark method. The filaments suitable for the blind-watermarking process have a melt viscosity between 4–24 kPa*s (PVA mixtures and EC mixtures). The filaments made of EVA have a very low viscosity of only 0.25 kPa*s, which was already suspected due to the non-existing bit structure in the printed EVA-tablets. The EVA + PVP-VA blend has the lowest viscosity, which is due to the high EVA content. Additionally, other excipients can reduce the viscosity, for example APIs or in this case mannitol. However, the EPO mixture also has a very low viscosity (0.169 kPa*s), although it was possible to detect the blind-watermarking patches made with these filaments. It seems that the property to solidify quickly after the polymer is melted has also a major influence on the blind-watermarking process and can compensate for the melt viscosity. Unfortunately, it was not possible to measure the solidification behavior over time. Therefore, this hypothesis cannot be confirmed here. An overview of the results of the blind-watermarking proof-of-concept is shown in Table 9.

Concluding, various influences can determine whether a material or mixture is suitable for the blind-watermarking process. Since the scanning process is very susceptible to light reflections, materials that are transparent or reflect light too strongly are not suitable, an issue that might be solved with another imaging technique. In addition, the quality of the blind-watermark code depends on the melting and solidification properties of the materials, so that only materials that have a high melt viscosity or very rapid solidification behavior can be considered. It must also be ensured that the filaments have a surface as homogeneous and smooth as possible, so that the inherent structure of the filaments does not interfere with the structure of the bit code. However, these influences can also be modified by adding a colorant or API that makes the filament appear less transparent and by changing the filament composition with components that increase melt viscosity and decrease solidification time.

## 4. Conclusions

In this study, a simple but powerful concept was tested to improve therapeutic safety and traceability of FDM 3D printed tablets for personalized medicine. It was shown that the blind-watermarking process can be used for oblong shaped tablets, as the bit-insertion and scanning process is currently only possible for straight, flat sides. Additionally, not every material is suitable for this approach. The color, roughness, and transparency of the filaments and printed objects have an impact on the detectability of the bits. However, when the filament used has a cloudy appearance or an added colorant, the detection process is feasible, and the bits are easy to detect. In addition, it is important to identify the correct print temperature of the used formulation since the formation of gas bubbles complicates the detection of the bits. For the formation of the bits, the melt viscosity and the solidification time of the printed filament seems to have a major impact. Unfortunately, a precise quantitative assessment of whether a filament is suitable for this process could not be made.

In comparison to other methods [39,42], this approach can be easily adapted in hospitals or community pharmacies with a cheap paper scanner and FDM 3D printer, as the dosage form and the blind-watermark codes are produced in one step without additional equipment. The method of Edinger et al. [42] requires a printer or film casting bench for the dosage form and an inkjet printer for the codes. Additionally, the traceability method of Trenfield et al. [39] needs more equipment for their track-and-trace process: a 2D and a 3D printer, for manufacturing the dosage form and printing the QR-code. Comparing the amount of information that can be put on a dosage form using these methods, more information can be covered in a QR code than in bits encoded in the tablet. Especially if the dosage form is very small, the information content per tablet is quite low. An attempt could be made to modify the bit encoding script, so that both sides of the tablet contain different information, thus doubling the bits per tablet. Nevertheless, 2^32^ variations can be encoded on a large tablet, resulting in almost 4.3 billion possible combinations. While this is not sufficient to replace a QR code, the simplicity of the approach allows the implementation of an additional layer of security without investments in further equipment.

In addition, other processes that build up the dosage form layer by layer and are controlled by G-Code could also be investigated (e.g., semi-solid 3D printing). Furthermore, it would be interesting to test other scan methods. It would be easier to use the camera of a smartphone with an application for detection. As most of the bits were easier to detect by eye than with the scanning procedure, it could extend the range of suitable materials and possibly the range of geometries, as round surfaces could be scanned.

## Figures and Tables

**Figure 1 pharmaceutics-14-00432-f001:**
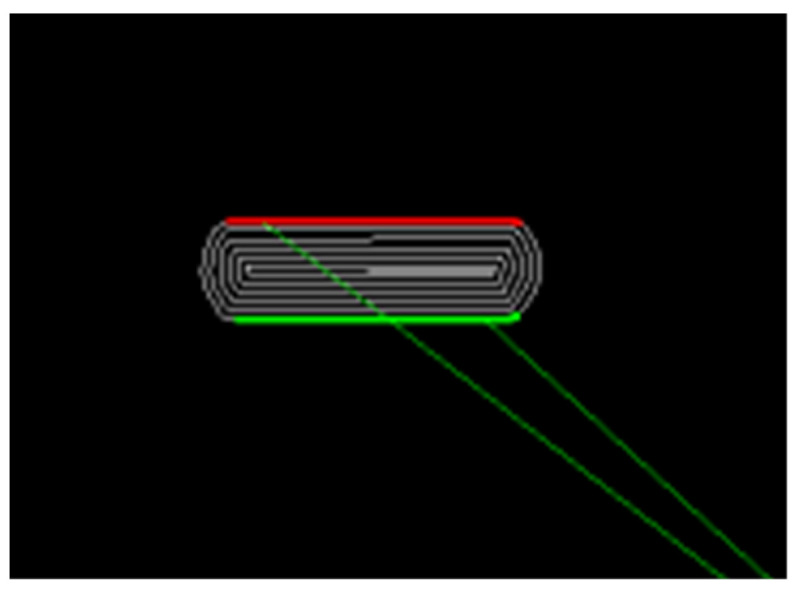
Detail from the watermark-embedding process. The side length (green + red) is recognized without the roundings and is encoded with bits.

**Figure 2 pharmaceutics-14-00432-f002:**
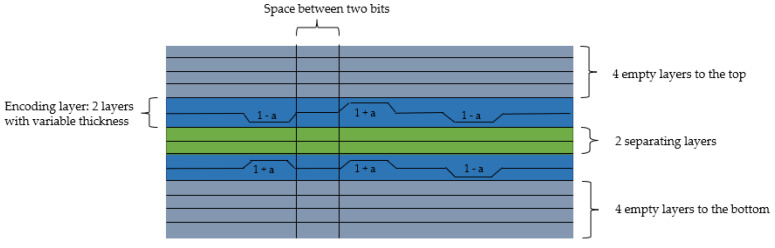
Scheme of the watermark embedding parameters and procedure. Adapted from the illustration in the original publication [53], IEEE, 2020.

**Figure 3 pharmaceutics-14-00432-f003:**
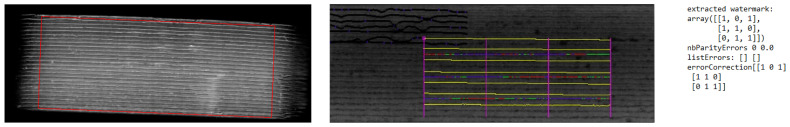
Example of the detection process (EC + HPMC); form left to right: scanned 3D printed oblong tablet with region of interest, detected watermark-patch and result of the bit detection.

**Figure 4 pharmaceutics-14-00432-f004:**
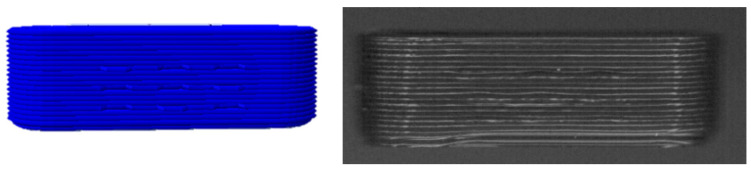
Pictured G-Code (**left**) and scanning image of 3D printed oblong tablet with 4 bits + 5 parity bits per side (**right**). Oblong tablet size: 12 mm length, 4 mm height, 4 mm width.

**Figure 5 pharmaceutics-14-00432-f005:**
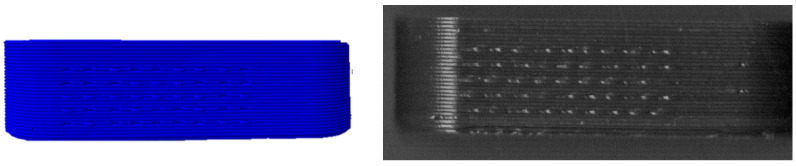
Pictured G-Code (**left**) and scanning image of 3D printed oblong tablet with 32 bits + 13 parity bits per side. Oblong tablet size: 23 mm length, 6 mm height, 8 mm width.

**Figure 6 pharmaceutics-14-00432-f006:**
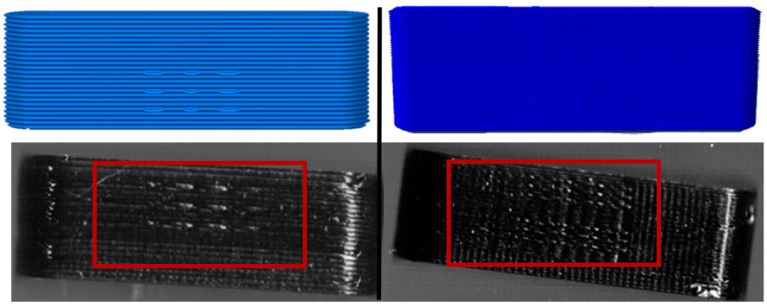
G-Codes (**top**) and scanning images of 3D printed oblong tablets (**bottom**). Left: 0.2 mm layer height (9 bits), right: 0.1 mm layer height (18 bits).

**Figure 7 pharmaceutics-14-00432-f007:**
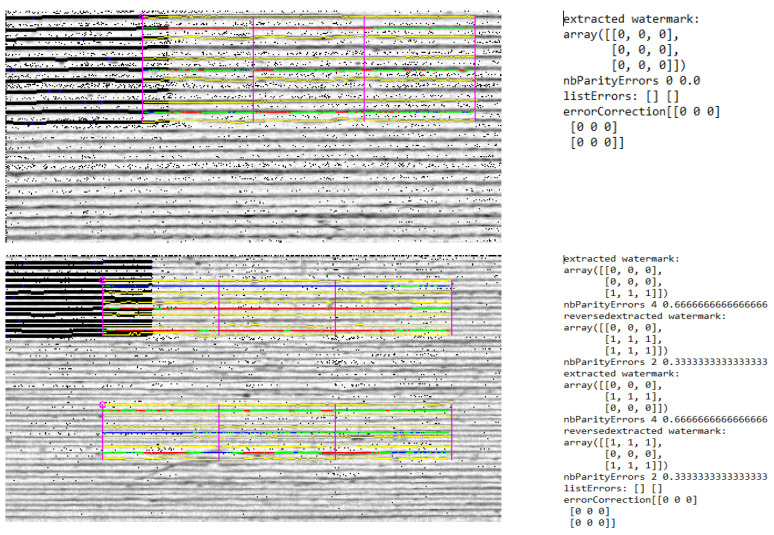
**Top**: results of the detection process of printed tablet with 0.2 mm layer height (9 bits). **Bottom**: Results of the detection process of printed tablet with 0.1 mm layer height (18 bits).

**Figure 8 pharmaceutics-14-00432-f008:**
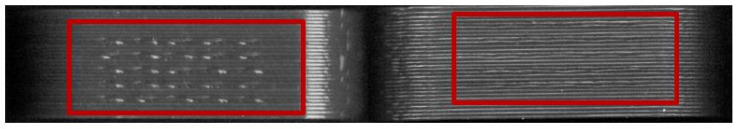
Influence of the orientation of the tablet with bits to the scan light. **Left**: perpendicular to the scan light, **right**: parallel to the scan light. Tablet: PLA with 25 bits.

**Figure 9 pharmaceutics-14-00432-f009:**
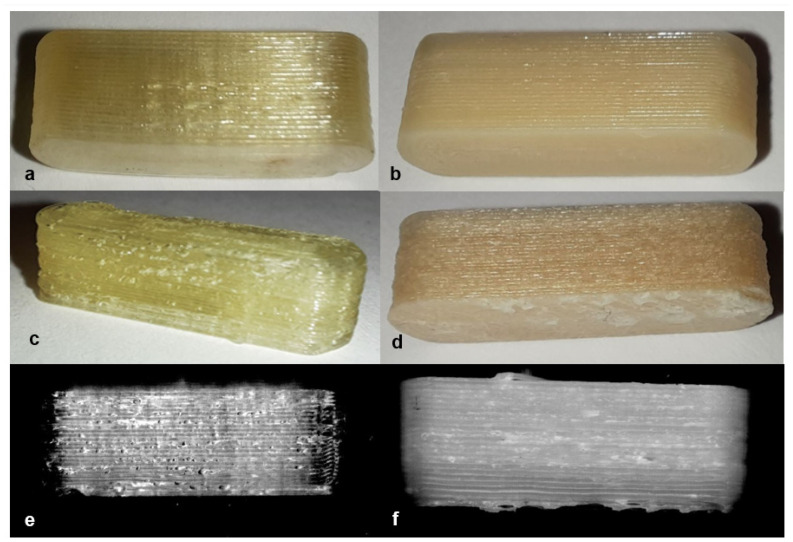
Filament PVA + PZQ printed at 188 °C (**a**) and 190 °C (**c**). Filament EPO + API printed at 176 °C (**b**) and 185 °C (**d**). Scanning image (**e**) present the scan of printed tablet shown in (**c**), and scanning image (**f**) present the scan of printed tablet shown in image (**d**).

**Table 1 pharmaceutics-14-00432-t001:** Composition of the filaments used.

Filament	Materials	Concentration/%	Manufacturer/Source
PLA	Polylactic acid (PLA)	100	Bavaria filaments, Freilassing, Germany
PVA	Polyvinyl alcohol (PVA)	100	Parteck MXP^®^, Merck, Darmstadt, Germany
PVA + PZQ [37]	Polyvinyl alcohol (PVA)	95	Parteck MXP^®^, Merck, Darmstadt, Germany
	Praziquantel (PZQ)	5	Bayer AG, Leverkusen, Germany
PVA + PDM [37]	Polyvinyl alcohol (PVA)	84	Parteck MXP^®^, Merck, Darmstadt, Germany
	Mannitol	10	Parteck M^®^, Merck, Darmstadt, Germany
	Pramipexole 2 HCl*H_2_O (PDM)	5	Chr. Olesen, Gentofte, Denmark
	Fumed silica	1	Aerosil^®^ 200 VV Pharma, Evonik, Essen, Germany
PVA + Triam [54]	Triamcinolone acetonide (Triam)	5	Farmabios, Gropello Cairoli, Italy
	Polyethylene glycol 300	10	Polyglycol 300, Clariant, Pratteln, Switzerland
	Polyvinyl alcohol (PVA)	85	Parteck MXP^®^, Merck, Darmstadt, Germany
PVA + colorant	Polyvinyl alcohol (PVA)	84	Parteck MXP^®^, Merck, Darmstadt, Germany
	Mannitol	10	Parteck M^®^, Merck, Darmstadt, Germany
	Methylene blue	5	Spectrum Lab Products, Gardena, CA, US
	Fumed silica	1	Aerosil^®^ 200 VV Pharma, Evonik, Essen, Germany
EPO + API	Basic butylated-methacrylate- copolymer (EPO)	80	Eudragit E PO^®^, Evonik, Essen, Germany
	Pramipexole 2 HCl*H_2_O (PDM)	20	Chr. Olesen, Gentofte, Denmark
EC	Ethyl cellulose (EC)	100	Aqualon^®^ N10, Ashland, KY, US
EC + HPMC [55]	Ethyl cellulose (EC)	72.93	Aqualon^®^ N10, Ashland, KY, US
	Hypromellose (HPMC)	16.67	Metolose 60SH 50, Shin Etsu Chemical, Tokyo, Japan
	Triethyl citrate	10	Citrofol AI Extra, Jungbunzlauer, Basel, Switzerland
	Fumed silica	0.4	Aerosil^®^ 200 VV Pharma, Evonik, Essen, Germany
EVA + PVA	Ethylene-vinyl acetate copolymer 82:18 (EVA)	25	Escorene^®^ FL01418, TER Chemicals, Hamburg, Germany
	Polyvinyl alcohol (PVA)	65	Parteck MXP^®^, Merck, Darmstadt, Germany
	Mannitol	10	Parteck M^®^, Merck, Darmstadt, Germany
EVA + PVP-VA + API	Ethylene-vinyl acetate copolymer 82:18 (EVA)	35	Escorene^®^ FL01418, TER Chemicals, Hamburg, Germany
	Vinylpyrrolidone-vinyl acetate copolymer 60:40 (PVP-VA)	15	Kollidon VA 64^®^, BASF, Ludwigshafen a. R., Germany
	Levodopa	40	Zhejiang Wild Wind Pharmaceutical, Dongyang, China
	Benserazide	10	BioPharma Synergies, Barcelona, Spain
ERL + API [56]	Anhydrous Theophylline	30	BASF, Ludwigshafen a. R., Germany
	Ammonio-methacrylate-copolymer type A (ERL)	62.6	Eudragit^®^ RL PO, Evonik, Essen, Germany
	Stearic acid	7	Baerlocher, Lingen, Germany
	Fumed silica	0.4	Aerosil^®^ 200 VV Pharma, Evonik, Essen, Germany

**Table 2 pharmaceutics-14-00432-t002:** Settings of the 3D printing process.

Filament	Bed Temperature/°C	Nozzle Temperature/°C
PLA	60	215
PVA	90	190
PVA + PZQ	90	188
PVA + PDM/colorant	60	188
PVA + Triam	60	190
EPO + API	45	176
ERL + API	55	180
EC	60	180
EC + HPMC	63	180
EVA + PVA	50	220
EVA + PVP-VA + API	50	220

**Table 3 pharmaceutics-14-00432-t003:** Settings of the scanning process.

Filament	Brightness	Contrast
PLA	−19	31
PVA	−100	30
PVA + PZQ	−100	55
PVA + PDM	−19	70
PVA + colorant	−50	25
PVA + Triam	−100	50
EPO + API	−100	60
ERL + API	−100	40
EC	−45	40
EC + HPMC	−80	55
EVA + PVA	−70	40
EVA + PVP-VA + API	−70	40

**Table 4 pharmaceutics-14-00432-t004:** Minimum bit insertion per side (parity bits marked in grey).

0	1	1
1	1	0
1	0	1

**Table 5 pharmaceutics-14-00432-t005:** Maximum bit code: 9 bits per line, 5 bits in height. 8 × 4 bits with 13 parity bits (grey marked).

0	0	1	0	0	1	1	1	0
0	0	0	1	1	1	0	0	1
1	1	1	1	1	1	1	0	1
1	0	0	1	0	0	1	0	1
0	1	0	1	0	1	1	1	1

**Table 6 pharmaceutics-14-00432-t006:** Bit code: 5 bits per line, 5 bits in height. 4 × 4 bits with 9 parity bits (grey marked).

1	1	0	0	0
0	1	1	0	0
1	1	0	1	1
1	1	0	0	0
1	0	1	1	1

**Table 7 pharmaceutics-14-00432-t007:** Material variations: images, scans and detection result.

Filament	Image	Scan (Parallel Orientation)	Detection	
PLA	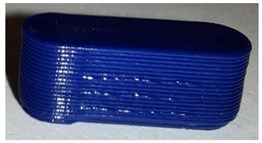	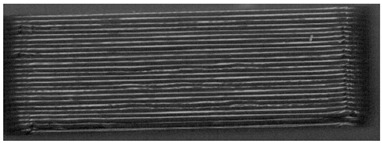	Yes 3/3	(a)
PVA	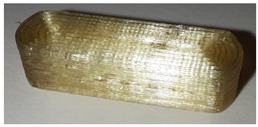	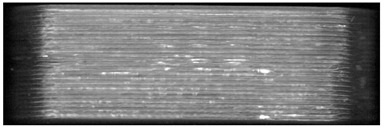	No 0/3	(b)
PVA + PZQ	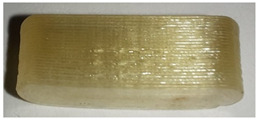	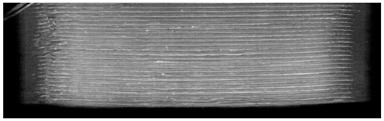	No 1/3	(c)
PVA + PDM	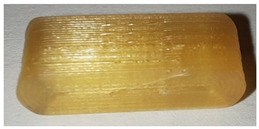	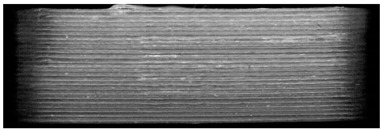	No 0/3	(d)
PVA + colorant	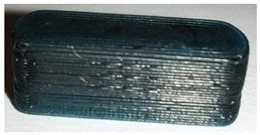	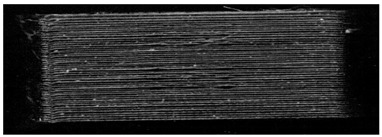	Yes 3/3	(e)
PVA + Triam	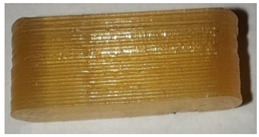	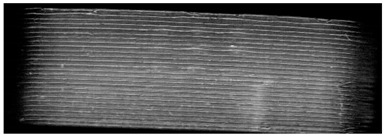	Yes 3/3	(f)
EPO + API	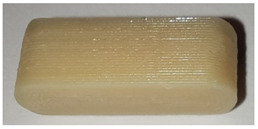	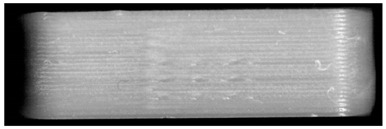	Yes 3/3	(g)
EC	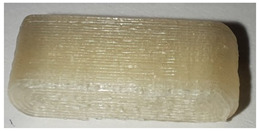	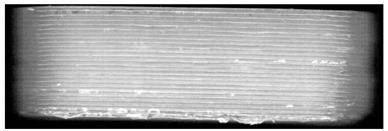	Yes 3/3	(h)
EC + HPMC	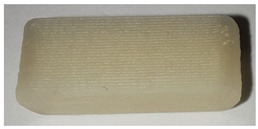	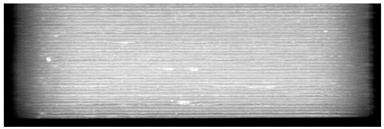	Yes 3/3	(i)
EVA + PVA	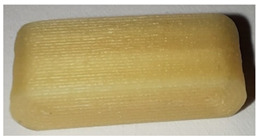	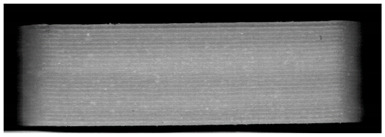	No 0/3	(j)
EVA + PVP -VA + API	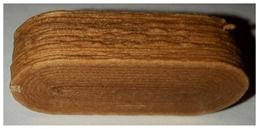	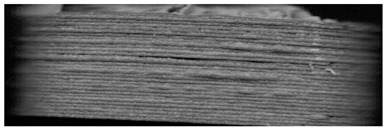	No 0/3	(k)
ERL + API	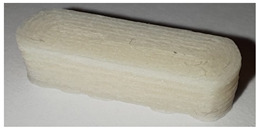	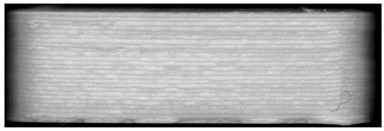	No 0/3	(l)

**Table 8 pharmaceutics-14-00432-t008:** Melt viscosity of the used materials at their specific print temperatures.

Filament	Nozzle Temperature/°C	Melt Viscosity/kPa*s
PVA	190	7.251
PVA + PZQ/PDM/colorant	188	6.407
PVA + Triam	190	4.321
EPO + API	176	0.169
EC	180	24.200
EC + HPMC	180	15.920
EVA + PVA	220	0.251
EVA + PVP-VA + API	220	0.135

**Table 9 pharmaceutics-14-00432-t009:** Overview of detection influences and results.

Filament	Transparent/ Reflective	Surface Roughness	Visible to the Eye	Detectable	Likely Reason
PLA	no	no	yes	yes	Good solidification behavior, no roughness, no reflection.
PVA	yes	no	yes	no	Transparent, reflection of the scan-light. High melt viscosity.
PVA + API	Dissolved API: yes Susp. API: no	no	yes	Dissolved API: no Susp. API: yes	Transparent, dissolved API does not decrease the reflection of the scan-light, suspended API or excipient forms slight milky filaments. High melt viscosity.
PVA + colorant	no	no	yes	yes	The colorant decreases the transparency of PVA. High melt viscosity.
EPO + API	no	no	yes	yes	Good solidification behaviour, low melt viscosity, no roughness, no reflection.
ERL + API	no	yes	no	no	Too rough, no bits recognizable.
EC	no	no	yes	yes	Good solidification behaviour, high melt viscosity, no roughness, no reflection.
EC + HPMC	no	no	yes	yes	Good solidification behaviour, high melt viscosity, no roughness, no reflection.
EVA + PVA	no	no	no	no	Solidification of the printed object occurs too slowly + low melt viscosity, the bits and layers deform.
EVA + PVP-VA + API	no	yes	no	no	Solidification of the printed object occurs too slowly + low melt viscosity, the bits and layers deform.

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
