# Peer review of "Blind-Watermarking—Proof-of-Concept of a Novel Approach to Ensure Batch Traceability for 3D Printed Tablets"

_pharmaceutics, 2022, doi:10.3390/pharmaceutics14020432_

Round 1

Reviewer 1 Report

Here, the authors integrate blind-watermarking process with fused deposition 3D printing technology to improve the therapeutic safety of the personalized medicine. They have found that color, roughness, and transparency of the printed structure have a significant impact on the readability of the bits. Clear structures make the detection of the bits challenging. Their study also shows that printing parameters such as temperature and flow speed can control the filament's roughness which can negatively impact the bits detection process. While this work might provide  some insight to the field, I believe too much of their process involves assumptions about their materials without ample evidence to support those assumptions.here are some comments and suggestions.

1-It has been claimed that the transparency of the printed filaments make the detection of the bits challenging. How about adding CaCO3 as a filler to the initial polymer? Would that improve the readability?

2-there is no information about the molecular weight of the polymers? how about their glass transition and melting temperature? any DSC thermogram?  how the processing temperature have been selected?

3-how about the other rheological properties? viscosity alone can not be sufficient to design and optimize the processing of polymers?

4- the roughness of the surface can also be due to the elastic instability and elastic fracture which can be controlled by choosing the right temperature and flow speed?

5- drying filaments before printing might prevent the bubble formation during the process. most of the processing machines are connected to a chamber in which polymer pellets are dried before entering the melting zone.)

6- EVA sample has a very low viscosity, is there any reason for choosing that particular sample. why not higher molecular weight?

7- It has been correctly mentioned that "the quality of the blind-watermark code depends on the melting and solidification properties of the materials",

however no data on the melting and solidification kinetics and properties have been provided.

Author Response

Here, the authors integrate blind-watermarking process with fused deposition 3D printing technology to improve the therapeutic safety of the personalized medicine. They have found that color, roughness, and transparency of the printed structure have a significant impact on the readability of the bits. Clear structures make the detection of the bits challenging. Their study also shows that printing parameters such as temperature and flow speed can control the filament's roughness which can negatively impact the bits detection process. While this work might provide some insight to the field, I believe too much of their process involves assumptions about their materials without ample evidence to support those assumptions. Here are some comments and suggestions.

 Thank you very much for your review and comments. We are happy to see that you are positive about our research even though you argue that several assumptions are not base on sufficient evidence.

While we describe several observations and phenomena, we also speculate what the underlying reasons might be. As this is a proof-of-concept study, it was not possible to perform systematic investigations to fully understand what the root causes are. We modified the text to make speculations clearer to the reader.

1-It has been claimed that the transparency of the printed filaments make the detection of the bits challenging. How about adding CaCO3 as a filler to the initial polymer? Would that improve the readability?

Thank you for this comment. It was observed that the code was more detectable on tablets that were not as transparent and had a more milky appearance. This can be achieved by the addition of colorants, as well as API and excipients, which do not dissolve or melt in the molten polymer-mixture, making the tablet opaque. Therefore, the addition of CaCO3 would likely improve the detectability of the bits, as it would not dissolve or melt in the polymer.

2-there is no information about the molecular weight of the polymers? how about their glass transition and melting temperature? any DSC thermogram?  how the processing temperature have been selected?

Thank you, this is an important point. DSC measurements were also made on the blends, but the print temperatures do not correspond to the melting ranges/ melting points/ TG of the polymers / blends. Print temperatures are above TG / MP due to very short dwell time in the printhead and very small nozzle diameter with 0.4 mm. The optimal print settings were determined manually. We start at lower temperatures and observe at which temperature a continuous filament flow comes out of the nozzle. From then on the temperature is tweaked until the best possible printing image is achieved. The EVA filaments were very flexible, so that the conveying wheels in the print head could not be used fully for the transport to the nozzle, otherwise the filament wrapped around the wheel after a few minutes. The necessary transport could only be guaranteed by a high printing temperature. In addition, the printing temperature had to be increased if the layers did not adhere to each other, which was the case with PVA-PZQ, PVA and EVA, for example. This description was added to chapter 2.2.4 for better understanding.

Unfortunately, there is no agreed upon and systematic approach to determine proper print settings that could be followed.

3-how about the other rheological properties? viscosity alone cannot be sufficient to design and optimize the processing of polymers?

 As written above, there is currently no common approach for the definition of FDM-process settings. In most published, pharmaceutical studies, not even the melt viscosity of the melt is known. It is simply investigated if the filament is printable and at what temperature a good quality is obtained. As simplistic as this is, it leads to high quality prints.

We realize that our findings highlight a need for a better understanding of the printing process. Application of creating of process models that take into account the viscosity, viscoelasticity, and cooling kinetics would be valuable but are not part of this study.

4- the roughness of the surface can also be due to the elastic instability and elastic fracture which can be controlled by choosing the right temperature and flow speed?

 The reviewer is right. The printing temperature has a very strong influence on the surface of the printed object. With a too high or too low temperature, fractures or even shark skinning can occur, which would make the surface appear uneven and destroy the bit structure. Therefore, the printing process for each filament was intensively examined and for each mixture the sweet spot was used for printing. However, despite the optimized printing temperature, it was not possible to compensate for the roughness in printing for all filaments.

 5- drying filaments before printing might prevent the bubble formation during the process. most of the processing machines are connected to a chamber in which polymer pellets are dried before entering the melting zone.)

That's correct, if the printing result is poor due to air bubbles formed by the hygroscopicity of the polymer, a solution to this can be found in the storage of the filaments. Under optimal (GMP) conditions, the filaments can be stored and also subsequently printed under controlled conditions, so that such influences can be eliminated or at least better controlled.

 6- EVA sample has a very low viscosity, is there any reason for choosing that particular sample. why not higher molecular weight?

 The EVA quality was selected for its release properties for a different study. Several filaments used in this study were selected because the formulations and processes were developed for other purposes. We deem this a suitable approach as formulation and process development for hot melt extrusion take a significant amount of time and we wanted to test this approach for multiple formulations.

Since EVA is a biodegradable polymer that does not dissolve in water and does not swell, it is very interesting for various issues, not only for oral dosage forms but also for printed implants. Blind watermarking is also interesting for these dosage forms. Two EVA variations were tested (VA28%/w and VA18%/w). Due to better printability and desired release properties, EVA with VA content 18 %/w was used.

 7- It has been correctly mentioned that "the quality of the blind-watermark code depends on the melting and solidification properties of the materials", however no data on the melting and solidification kinetics and properties have been provided.

Thank you for this critical analysis. With the tools available, we tried to quantify the influencing parameters as well as possible. Unfortunately, one strongly influencing parameter is the solidification behavior, which is not measurable with the equipment at hand.

The melt properties were investigated with regard to dynamic viscosity, since the flow behavior out of the nozzle and the internal stability of the printed strand contribute to maintaining the watermark pattern. Since the investigations regarding the solidification behavior could not be carried out, this hypothesis of the influence is clarified better.

Reviewer 2 Report

The authors dealt to provide a proof for the applicability of a previously published anti-counterfeit concept. The selected topic has high importance, and the paper contains significant novelty in this field.

line 33: please write developing countries instead of development countries

The paper is well written, and the conclusions are well supported by the results. Nevertheless, there are some minor concerns which should be addressed before publication:

The description of the watermark embedding procedure should be clearer, it should be better described that the encoding layer is consisted of two individual layers and if the bit is 1 the bottom layer thickness is 1+a and the top layer is 1-a to keep the thickness of the two layers constant. In the current form it was a bit hard to understand.

Figure 9 the order of the figures should be d, b, c, a, e, f so the pictures of the same composition would appear below each other and the quality change could be followed easier

Table 6 could be deleted since it equals with table 4. You can mention that the same coding was used.

In case of Figure 8 it would be good to know the encoding matrix, in order to estimate if the brightness of spots of the embedded bits are in accordance of 0 or 1 bit value or not, because if yes the perpendicular scanning may be the solution for those samples where reading of the code was impossible with parallel scanning.

Author Response

The authors dealt to provide a proof for the applicability of a previously published anti-counterfeit concept. The selected topic has high importance, and the paper contains significant novelty in this field.

We thank the reviewer for his efforts and positive opinion about our submission. Even though our approach does not allow complete traceability, it is capable of adding another layer of security to printed medicines without many compromises.

 Line 33: please write developing countries instead of development countries

Thanks for the thoughtful review, the error has been corrected in the manuscript.

The paper is well written, and the conclusions are well supported by the results. Nevertheless, there are some minor concerns which should be addressed before publication:

The description of the watermark embedding procedure should be clearer, it should be better described that the encoding layer is consisted of two individual layers and if the bit is 1 the bottom layer thickness is 1+a and the top layer is 1-a to keep the thickness of the two layers constant. In the current form it was a bit hard to understand.

 Thanks for the advice. The section 2.2.3 about embedding the blind-watermark code on the printed tablet has been revised again and described in more detail.

 Figure 9 the order of the figures should be d, b, c, a, e, f so the pictures of the same composition would appear below each other and the quality change could be followed easier

You're absolutely right, it's much clearer in that order. The figure has been adjusted.

Table 6 could be deleted since it equals with table 4. You can mention that the same coding was used.

 Thank you, you are right. The table was deleted and referred to the previous table.

 In case of Figure 8 it would be good to know the encoding matrix, in order to estimate if the brightness of spots of the embedded bits are in accordance of 0 or 1 bit value or not, because if yes the perpendicular scanning may be the solution for those samples where reading of the code was impossible with parallel scanning.

Thanks for the comment, a table with the code used has been added (Table 6). Unfortunately, the scanner does not recognize the perpendicular orientation of the bits well. All tablets were scanned in both orientations and evaluated for detection. The script detects the bits better when the layer structure can be seen, even if the bits are worse for the viewer to see in these images.

Round 2

Reviewer 1 Report

no comments